# A Bayesian Method for Dynamic Origin–Destination Demand Estimation Synthesizing Multiple Sources of Data

**DOI:** 10.3390/s21154971

**Published:** 2021-07-21

**Authors:** Hang Yu, Senlai Zhu, Jie Yang, Yuntao Guo, Tianpei Tang

**Affiliations:** 1School of Transportation and Civil Engineering, Nantong University, Se Yuan Road #9, Nantong 226019, China; yuhang@ntu.edu.cn (H.Y.); yangjie110@ntu.edu.cn (J.Y.); tangtianpei@ntu.edu.cn (T.T.); 2Key Laboratory of Road and Traffic Engineering, Ministry of Education, Department of Traffic Engineering Tongji University, 4800 Cao’an Road, Shanghai 201804, China; yuntaoguo@tongji.edu.cn

**Keywords:** dynamic O–D estimation, Bayesian statistic, synthesizing data, stepwise algorithm

## Abstract

In this paper a Bayesian method is proposed to estimate dynamic origin–destination (O–D) demand. The proposed method can synthesize multiple sources of data collected by various sensors, including link counts, turning movements at intersections, flows, and travel times on partial paths. Time-dependent demand for each O–D pair at each departure time is assumed to satisfy the normal distribution. The connections among multiple sources of field data and O–D demands for all departure times are established by their variance-covariance matrices. Given the prior distribution of dynamic O–D demands, the posterior distribution is developed by updating the traffic count information. Then, based on the posterior distribution, both point estimation and the corresponding confidence intervals of O–D demand variables are estimated. Further, a stepwise algorithm that can avoid matrix inversion, in which traffic counts are updated one by one, is proposed. Finally, a numerical example is conducted on Nguyen–Dupuis network to demonstrate the effectiveness of the proposed Bayesian method and solution algorithm. Results show that the total O–D variance is decreasing with each added traffic count, implying that updating traffic counts reduces O–D demand uncertainty. Using the proposed method, both total error and source-specific errors between estimated and observed traffic counts decrease by iteration. Specifically, using 52 multiple sources of traffic counts, the relative errors of almost 50% traffic counts are less than 5%, the relative errors of 85% traffic counts are less than 10%, the total error between the estimated and “true” O–D demands is relatively small, and the O–D demand estimation accuracy can be improved by using more traffic counts. It concludes that the proposed Bayesian method can effectively synthesize multiple sources of data and estimate dynamic O–D demands with fine accuracy.

## 1. Introduction

Intelligent transportation systems (ITS) have been vigorously implemented in many cities around the world. The effectiveness of real-time traffic management strategies of ITS normally depends on reliable dynamic (i.e., time-dependent) origin–destination (O–D) demand estimation. In general, O–D demand matrices can be obtained either from household surveys or/and estimated by traffic counts. O–D demand obtained by household surveys is not only costly but also vulnerable to become outdated. Thus, the use of traffic counts to estimate O–D demand becomes attractive because it is cheap and easy to collect data and to implement.

There is a rich body of literature estimating static or dynamic O–D demand using link counts. In real applications, the number of links is usually far less than the number of O–D pairs, thus the O–D demand estimation problem based on link counts becomes an underspecified (or degenerate) problem that has no unique solution. Therefore, additional information is required to acquire a unique solution. In ITS, various types of sensors (GPS, blue tooth, video, automatic vehicle identification (AVI), plate scanning, etc.) are adopted to collect traffic data and to predict traffic state. By these sensors, link counts, intersection turning movements, flows, and travel time on links and partial paths can be collected.

According to the type of traffic data collected, sensors in this paper are categorized as follows. (a) Counting sensors: these sensors include inductive loop detectors, magnetic detectors, etc. Using these sensors, traffic characteristics such as speed, density, occupancy, and flow rates on a single lane or a set of lanes in the network can be measured. (b) Image/video sensors: based on these sensors, moving flows at an intersection or a link can be collected by taking images or videos. (c) Vehicle-ID sensors: these sensors include license plate readers, GPS, etc. By these sensors, vehicle IDs in the network can be identified, thus full or partial path-related information of vehicle can be inferred. For instance, GPS can be deployed on vehicles to track their route.

Although various sources of data can be used in O–D demand estimation, most studies tend to use just one source of data or combine it with link counts; very few studies synthesize multiple sources of data to estimate O–D demand. Synthesizing multiple sources of data is difficult because they are correlative and complementary to each other and thus cannot be simply combined together. Hence, the statistical correlations among them should be analyzed. Moreover, most studies do not make use of travel times (travel times on links or partial paths) in O–D estimation. This is because the connections between travel time and O–D demand cannot be measured directly. However, travel time data can often be collected much more easily (e.g., by vehicle-ID sensors) than the volume data (especially along a path or a partial path). For example, travel time of a path or a partial path can be derived by tracking the departure time and arrival time of several GPS-equipped vehicles. However, it is hard to get path traffic volume data since all the vehicles using the same path are difficult to track synchronously. Therefore, it is worthwhile to consider the travel time data in O–D demand estimation.

To bridge the gap above, this paper tries to synthesize multiple sources of data together, mainly including link counts, time-varying flows and travel time along partial observed paths, and turning movements at intersections, to estimate dynamic O–D demand. Specifically, we treat O–D demands as random variables satisfying multivariate normal distribution, and propose a Bayesian statistical model to estimate dynamic O–D demand by synthesizing these multiple sources of data. By solving the dynamic user equilibrium (DUE) problem based on an assumed prior O–D demand, the prior distribution (including a vector of expected values and a variance-covariance matrix) of all considered variables is estimated. The relationships among all variables are analyzed by variance-covariance matrices. By updating the assumed prior distribution of all variables using traffic counts, we establish the posterior distributions of all variables, based on which point estimation and probability confidence intervals are inferred to measure the intrinsic uncertainty. In the proposed Bayesian statistical model, we convert the observed sub-path travel time to several sub-path flows so as to incorporate sub-path travel time information in O–D estimation. Specifically, for a sub-path with a given departure time, we sample the normal distributed sub-path travel time to get arrival time for each user, and the mean of all the normal distributed sub-path travel times is equal to the observed sub-path travel time. By this sampling method we convert the sub-path travel time information to sub-path flows which is more appropriately analyzed in O–D estimation.

One challenge in solving the Bayesian statistical model is to update the traffic counts and calculate the posterior distribution of all variables, since it involves many matrix inversions during the calculation, especially on large-scale networks. To simplify the calculation, a stepwise algorithm that avoids matrix inversions is introduced to solve the proposed Bayesian statistical model. In this algorithm, the posterior distribution is estimated by sequentially updating traffic count one by one. In this process, matrix inversions are not needed since all matrices in the model degenerate to vectors or scalars.

The remainder of the paper is organized as follows. In Section 2, the related literature of O–D demand estimation methods are reviewed. Then, the Bayesian statistical method is briefly explained in Section 3. Section 4 proposes a Bayesian statistical model for dynamic O–D estimation, followed by a stepwise algorithm for solving the Bayesian model in Section 5. Numerical examples are provided in Section 6 to illustrate the proposed models and algorithm. Finally, some conclusions are given in Section 7.

## 2. Literature Review

### 2.1. Static O–D Demand Estimation

O–D demand estimation was studied extensively for static case. These methods can be classified as:

(1) Least squares [1,2] and generalized least squares (GLS) [3,4,5] methods. These methods are usually bi-level problems. The upper level is to minimize the weighted distances between the target and estimated O–D demands, and/or between the traffic counts and estimated traffic volumes; the lower level model is a static user equilibrium problem.

(2) Entropy concept-based methods [6]. The entropy concept measures the reasonableness and closeness to reality of an estimated O–D demand matrix. Subject to the prior O–D matrix, the probability distribution of O–D demand which best represents the current state of knowledge is the one with the maximum entropy.

(3) Maximum likelihood methods [7]. Assuming the elements of the prior O–D demand matrix are obtained from random variables with given probability distribution, these methods maximize the likelihood of the traffic counts conditional on the estimated O–D demand matrix and the prior O–D demand matrix.

(4) Bayesian inference and Bayesian network methods [8,9,10,11,12,13,14]. These methods treat traffic flow as random variables. Using observed traffic counts to update the assumed prior distribution, the posterior distribution of all variables is built based on Bayes’ theorem.

Overall, in order to obtain a unique solution, these methods usually use a prior matrix (or seed matrix). Since the accuracy of the O–D demand estimation is affected significantly by the prior matrix, some researchers proposed methods based on traffic counts only to estimate O–D demand [15,16].

### 2.2. Dynamic O–D Demand Estimation

Estimating dynamic O–D demand is more complicated than the static O–D case due to its time-varying characteristic. Some studies straightforwardly extend methods for static case to the dynamic case using time-dependent link counts. For example, to estimate dynamic O–D demand, Cascetta et al. [17] proposed a GLS method based on a simplified assignment model. Following Cascetta et al. [16], several researchers extended CLS methods for dynamic O–D demand estimation [18,19,20]. For example, Guo et al. [20] proposed a least square method to estimate dynamic O–D matrix using radio frequency identification data. In addition, state space models are also frequently used based on the state vector indicating the unknown O–D demand [21,22,23].

Note that most existing methods for dynamic O–D demand estimation problem are characterized by a bi-level optimization structure. The upper-level problem is to minimize two deviation functions: (1) the distances between observed and estimated time-dependent traffic counts and (2) the distances between the prior and estimated dynamic O–D demand matrices. The lower-level problem is the dynamic user equilibrium (DUE) problem. To solve such a bi-level optimization problem, researchers proposed various algorithms/methods, such as advanced parallel evolutionary algorithm [24], gradient approximation method [25,26], guided genetic algorithm [27], cluster-wise simultaneous perturbation stochastic approximation algorithm [28], etc.

Meanwhile, single-level formulations have also been proposed for the dynamic O–D demand estimation problem. For instance, based on variational inequalities (VI), Nie and Zhang [29] formulated a single-level formulation utilizing the dynamic link-path incidence relationships in a generic projection-based VI solution framework. Lundgren and Peterson [30] proposed a heuristic algorithm by adapting the projected gradient method based on a single level reformulation.

### 2.3. O–D Demand Estimation Using Multiple Sources of Data

Due to the development in real-time sensing technologies, a variety of sensors can be used to collect different types of traffic data, including time-dependent link flows, turning movements at intersections, and full or partial path flows and path travel time.

Turning movements at intersections are normally detected by image/video sensors. Intersection turning movements provide more information on users’ travel behavior and usage of network topologies than link counts. Several studies estimate O–D demand by using turning movements at intersections. For instance, Yang et al. [31] proposed a neural network approach to estimate dynamic O–D demand using node-based traffic counts. Using both turning movements and link counts, Alibabai and Mahmassani [32] presented a bi-level optimization method for dynamic O–D demand estimation, and Lu et al. [33] proposed a Kalman filter approach to estimate dynamic O–D demand.

To estimate O–D demand, path-based information is more desirable because it can fully reflect users’ route choice behavior and network topology. However, path information cannot be fully detected, so researchers normally make use of observed flows or data on partial path, which can be captured by GPS, mobile phone, plate scanning, AVI, etc. [34,35,36,37,38,39]. For instance, to estimate static O–D demand, Hu et al. [35] proposed link-based and path-based models to estimate O–D demand based on traffic counts by vehicle detector sensors and license plate recognition. In the dynamic case, Yang et al. [37] proposed two GLS models formulated as single-level optimization problems based on both probe vehicle trajectories and link counts. Krishnakumari et al. [38] proposed a method without dynamic network loading using measured flows and speeds. Cao et al. [39] estimated day-to-day dynamic O–D demand based on connected vehicle trajectories and automatic vehicle identification data.

In real application, observing partial path or sub-path flow could be difficult and costly. However, path-based travel time can be observed much more easily and thus can be used in O–D estimation. For example, Dixon and Rilett [40] applied the GLS method proposed by Cascetta et al. [17] to estimate the link-flow proportions based on the observed travel time. Based on local link marginal travel time evaluation by adapting the method proposed by Ghali and Smith [41], Qian and Zhang [42] extended the single-level O–D demand estimation framework proposed by Nie and Zhang [3] to utilize travel time measurements.

Except for the above measurements, O–D demand is also estimated by using other types of traffic information. For instance, since speed and density provide the best representation of traffic congestion, some researchers made use of these traffic measures to estimate dynamic O–D demand [43,44,45]. Recently, mobile phone data are used by some researchers to infer O–D trips [46,47,48].

In the literature, although various sources of data can be used in O–D demand estimation, as shown in Table 1, most studies tend to use just one source of data, or combine it with link counts; very few studies synthesize multiple sources of data together to estimate O–D demand. In addition, travel times (travel times on links or partial paths) are rarely used in O–D estimation. Statistical methods which have been frequently used in static O–D estimation are also rarely adopted in dynamic case. Providing variability information of the traffic flow estimation is the most important advantage of statistical methods. Normally, other methods give only the particular values of the O–D and link flows, while statistical methods could also provide the corresponding probability intervals. Although variability information is important in real applications, it is difficult to capture this information in dynamic O–D demand estimation due to the time-varying characteristic. In Zhu et al. [49], heterogeneous sensor deployment strategies were proposed for dynamic O–D demand estimation. Based on the Bayesian method adopted in Zhu et al. [49], this paper tries to synthesize multiple sources of data, mainly including link counts, time-varying flows and travel time along partial observed paths, and turning movements at intersections, to estimate dynamic O–D demand. Variability information of the traffic flow estimation can also be provided by using the proposed Bayesian method.

## 3. Bayesian Statistical Method

In this section, some background of Bayesian statistical method are illustrated and how the information can be updated using the Bayesian statistical method is described.

The Bayesian statistical method is a suitable method for combining prior (historical) information and sample information. Bayesian inference and Bayesian network methods have been used frequently to solve a wide variety of practical problems [50,51,52,53,54,55]. These methods have also been applied widely to solve the O–D demand estimation problem [8,9,10,11,12,13,14,16]. In a Bayesian statistical method, variables are not fixed; rather, they are random variables satisfying given probability distributions. A Bayesian statistical method usually updates the prior distribution by using sample information in order to obtain the posterior distribution p(θ|X), based on Bayes theorem as follows:(1)$$p\left(\theta |X\right)=\frac{f\left(X|\theta \right)p\left(\theta \right)}{\int \;f\left(X|\theta \right)p\left(\theta \right)d\theta }$$
where p(θ) specifies the prior probability density function of parameter **θ**. In this paper **θ** refers to a set of random variables, including dynamic O–D demand, time-dependent path and sub-path flows, turning movements at intersections and link flows. In Equation (1), f(X|θ) is the likelihood of observation X, including partial observed time-varying sub-path flows, turning movements at intersection, and link counts; ∫f(X|θ)p(θ)dθ is simply the marginal density of X, which does not depend on the value of θ.

Once p(θ|X) is identified, we can obtain the point estimation $\tilde{\theta }$ by solving the following maximum posterior density planning:(2)$$\tilde{\theta }=\arg\;\underset{\theta }{\max}p\left(\theta |X\right)$$

In a transportation network with modest size, it is difficult to calculate the posterior distribution due to the multidimensional integral over the feasibility domain. Normally, Metropolis–Hastings algorithms are used to calculate the posterior distribution [10,12]. These algorithms normally need a large number of samples and are heuristic methods.

Because traffic flows are treated as random variables in the Bayesian statistical method, we make the following assumption on their distributions:

**Assumption.** The time-dependent traffic demands between all O–D pairs are assumed to follow multivariate normal (MVN) distributions.

Previous studies [56,57] made similar assumptions. Note that a normal distribution for traffic flows is reasonable, because their probabilities can be treated as success rate among repeating a large number of independent Bernoulli experiments in which the users randomly make travel decisions. Moreover, the prior and posterior distributions are conjugate in the case of multivariate normal distribution. If the posterior distribution is in the same family as the prior probability distribution, they are called conjugate distributions. A Bayesian statistical method often works with conjugate priors such that their associated posteriors belong to the same families. It is noteworthy that the normal distribution allows negative travel demands in theory, which is not realistic. However, such an issue could be mitigated if the model is calibrated carefully to fit the observed data well.

Specifically, if θ is normally distributed with mean $\mu_{\theta }$ and variance-covariance $\Sigma_{\theta }$, calculating the posterior distribution p(θ|X) is to calculate the posterior mean $\mu_{\theta |X=x}$ and variance-covariance $\Sigma_{\theta |X=x}$, which can be obtained by using the following updating equations [8,14]:(3)$$\mu_{\theta |X=x}=\mu_{\theta }+\Sigma_{\theta X}\Sigma_{XX}^{-1}\left(x-\mu_{X}\right)$$
(4)$$\Sigma_{YZ|X=x}=\Sigma_{YZ}-\Sigma_{YX}\Sigma_{XX}^{-1}\Sigma_{XZ}$$
where Y and Z both refer to the components of θ; $\mu_{X}$ and $\Sigma_{XX}$ are the mean vector and variance-covariance matrix of observation X; $\Sigma_{\theta X}$ is the variance-covariance matrix of θ and X; $\Sigma_{YZ}$ is the variance-covariance matrix of Y and Z; $\Sigma_{YX}$ is the variance-covariance matrix of Y and X; $\Sigma_{XZ}$ is the variance-covariance matrix of X and Z; and x is the actual observed value of X. Note that the posterior mean $\mu_{\theta |X=x}$ depends on x, but the posterior variance-covariance $\Sigma_{YZ|X=x}$ does not.

In addition, in case of multivariate normal distribution, the posterior mean $\mu_{\theta |X=x}$ is equal to the optimal $\tilde{\theta }$ of the maximum posterior density planning (2); that is, we can take the posterior mean $\mu_{\theta |X=x}$ as the point estimation of θ.

## 4. Model Formulation

In this section, we derive the formulation of the Bayesian statistical model for the dynamic O–D estimation. Since multiple sources of data are considered, we first analyze relationships among traffic flows between different sources of data.

### 4.1. Relationships among Time-Dependent Traffic Flows

#### 4.1.1. O–D Demand and Path Flow

Denote $d_{i,t}$ as the flow of O–D pair i at departure time t, $f_{i,k,t}$ as the number of users between O–D pair i choosing path k at departure time t, and $p_{i,k,t}$ as the proportion of users between O–D pair i at departure time t choosing path k.

According to the conservation law, $f_{i,k,t}$ can be obtained by:(5)$$f_{i,k,t}=p_{i,k,t}d_{i,t}$$

Define vectors D, $F_{i}$, F as follow:(6)$$D={\left[d_{1,1},d_{1,2},\ldots ,d_{1,t},\ldots ,\underset{\text{time-dependent demand of O–D}\;i}{\underset{︸}{d_{i,1},d_{i,2}\ldots ,d_{i,t}\ldots }},\ldots \right]}^{T}$$
(7)$$F_{i}={\left[f_{i,1,1},f_{i,1,2},\ldots ,f_{i,1,t},\ldots ,\underset{\text{time-dependent flow of path k from O–D}\;i}{\underset{︸}{f_{i,k,1},f_{i,k,2},\ldots ,f_{i,k,t}\ldots }},\ldots \right]}^{T}$$
(8)$$F={\left[F_{1}^{T},F_{2}^{T},\ldots ,F_{i}^{T},\ldots \right]}^{T}$$
where D is the vector consists of all considered time-dependent O–D demand, $F_{i}$ is the vector consists of all time-dependent path flows between O–D pair i, and F is the vector consists of all time-dependent path flows.

Define a m×s matrix $P_{i,t}$, where m refers to the number of paths for O–D pair i at departure time t, s equals the dimension of **D**, then the (*k*, *j*) element $P_{i,t}\left(k,j\right)$ of $P_{i,t}$ is defined by:(9)$$P_{i,t}\left(k,j\right)=\{\begin{array}{cc} p_{i,k,t}, & if\;j=\left(i-1\right)*\left|T\right|+t\\ 0, & otherwise \end{array}$$
where |T| denotes the number of time intervals.

Define matrix P as follows:(10)$$P={\left[P_{1,1}^{T},P_{1,2}^{T},\ldots ,P_{2,1}^{T},P_{2,2}^{T},\ldots ,P_{i,1}^{T},P_{i,2}^{T},\ldots \right]}^{T}$$

Then the time-dependent path flows satisfy the following flow conservation condition:(11)F=PD

#### 4.1.2. O–D Demand, Path Flow, and Sub-Path Flow

Define $f_{sub,j,\tau_{a},\tau_{d}}$ as the flow along sub-path j with departure time $\tau_{d}$ at the start node and arrival time $\tau_{a}$ at the tail node, $\varphi_{i,k,t,j,\tau_{a},\tau_{d}}$ as the proportion of users between path k of O–D pair i with departure time t choosing sub-path j with arrival time $\tau_{a}$ and departure time $\tau_{d}$.

The time-dependent sub-path flow can be derived from the time-dependent path flow as follows:(12)$$f_{sub,j,\tau_{a},\tau_{d}}=\underset{i}{\sum }\underset{k}{\sum }\underset{t}{\sum }\varphi_{i,k,t,j,\tau_{a},\tau_{d}}f_{i,k,t}$$

Define vectors $\varphi_{i,j,\tau_{a},\tau_{d}}$ $\varphi_{j,\tau_{a},\tau_{d}}$, φ and $F_{sub}$ as follows:(13)$$F_{sub}={\left[f_{sub,1,1,1},f_{sub,1,2,1},\ldots ,\underset{\text{time-dependent flow of sub-path j}}{\underset{︸}{f_{sub,j,1,1},f_{sub,j,2,1},\ldots f_{sub,j,\tau_{a},\tau_{d}}\ldots }},\ldots \right]}^{T}$$
(14)$$\varphi_{i,j,\tau_{a},\tau_{d}}=\left[\varphi_{i,1,1,j,\tau_{a},\tau_{d}},\varphi_{i,1,2,j,\tau_{a},\tau_{d}},\ldots ,\varphi_{i,k,1,j,\tau_{a},\tau_{d}},\varphi_{i,k,2,j,\tau_{a},\tau_{d}},\ldots ,\varphi_{i,k,t,j,\tau_{a},\tau_{d}},\ldots \right]$$
(15)$$\varphi_{j,\tau_{a},\tau_{d}}=\left[\varphi_{1,j,\tau_{a},\tau_{d}},\varphi_{2,j,\tau_{a},\tau_{d}},\ldots ,\varphi_{i,j,\tau_{a},\tau_{d}},\ldots \right]$$
(16)$$\varphi ={\left[\varphi_{1,1,1}^{T},\varphi_{1,2,1}^{T},\ldots ,\varphi_{j,1,1}^{T},\varphi_{j,2,1}^{T},\ldots \right]}^{T}$$
where $F_{sub}$ is the vector consists of all considered time-dependent sub-path flow and φ is the corresponding proportion vector. $\varphi_{i,j,\tau_{a},\tau_{d}}$ is the vector consists of the proportions of users between all time-dependent paths of O–D pair i choosing sub-path j with arrival time $\tau_{a}$ and departure time $\tau_{d}$. $\varphi_{j,\tau_{a},\tau_{d}}$ is the vector consists of the proportions of users between all time-dependent paths of all O–D pairs choosing sub-path j with arrival time $\tau_{a}$ and departure time $\tau_{d}$.

Then consider the error term; a linear relationship between the O–D demand vector D, the path flow vector F and the sub-path flow vector $F_{sub}$ is assumed to be:(17)$$F_{sub}=\varphi F+\varepsilon =\varphi PD+\varepsilon$$
where $\varepsilon =\left(\varepsilon_{1},\varepsilon_{2},\ldots \right)$ are mutually independent random variables with zero mean.

However, sometimes we can only observe the time-dependent sub-path travel time rather than the sub-path flow. In order to make use of the observed sub-path travel time, we convert these observed travel times to time-dependent sub-path flows.

Denote ${\bar{t}}_{j,\tau_{d}}$ as the observed mean travel time of sub-path j with departure time $\tau_{d}$ at the start node. Then for sub-path j with departure time $\tau_{d}$, the random travel time $t_{j,\tau_{d}}$ is expressed as:(18)$$t_{j,\tau_{d}}={\bar{t}}_{j,\tau_{d}}+\gamma_{j,\tau_{d}}$$
where $\gamma_{j,\tau_{d}}$ is the random term and is assumed to be normal distributed with zero mean and variance $\lambda {\bar{t}}_{j,\tau_{d}}$, where λ is the coefficient of variation.

Since we cannot obtain the true time-dependent sub-path flow, we assume that the total flow of sub-path j with departure time $\tau_{d}$ is equal to the prior (or historical) flow, which can be obtained by solving the dynamic user equilibrium problem based on the prior time-dependent O–D demand. For each user, since the travel time is random, it can be obtained by random sampling from the normal distribution $N\left({\bar{t}}_{j,\tau_{d}},\lambda {\bar{t}}_{j,\tau_{d}}\right)$. Finally, by sorting out the travel time of all users, we can derive the flows of sub-path j with different arrival time, which can be treated as the observed time-dependent sub-path flows.

#### 4.1.3. O–D Demand, Path Flow, and Intersection Turning Movements

Each node (intersection) turning movement can be treated as a sub-path with three nodes: the upstream node, the intersection node, and the downstream node. The arrival time and departure time at the upstream and downstream nodes cannot be observed. The departure time at the considered intersection can be inferred as follows.

Define $s_{j_{a,b},\tau }$ as the number of users traveling from upstream node a to downstream node b connected by node j with departure time τ at node j, and $\phi_{i,k,t,j_{a,b},\tau }\;$ as the proportion of users between path k of O–D pair i with departure time t at the origin node of O–D pair i traveling from upstream node a to downstream node b connected by node j at time τ, where $a\in N_{u}$ and $b\in N_{d}$, $N_{u}$ is the set of upstream nodes of node j and $N_{d}$ is the set of downstream nodes of node j.

The time-dependent turning movement can be derived from the time-dependent path flow as follows:(19)$$s_{j_{a,b},\tau }=\underset{i}{\sum }\underset{k}{\sum }\underset{t}{\sum }\phi_{i,k,t,j_{a,b},\tau }f_{i,k,t}$$

Define a column vector $S_{j,\tau }$ as the set of turning movements at node j with departure time τ and a row vector $\phi_{i,k,t,j,\tau }$ as the set of the proportions of users between path k with departure time t of O–D pair i choosing the corresponding turning movement at node j with departure time τ. Then vectors $\phi_{i,j,\tau }$, $\phi_{j,\tau }$, ϕ and S are expressed as follows:(20)$$S={\left[S_{1,1},S_{1,2},\ldots \underset{\text{time-dependent turning movements at node j}}{\underset{︸}{S_{j,1},S_{j,2},\ldots ,S_{j,\tau },\ldots }},\ldots \right]}^{T}$$
(21)$$\phi_{i,j,\tau }=\left[\phi_{i,1,1,j,\tau },\phi_{i,1,2,j,\tau },\ldots ,\phi_{i,k,1,j,\tau },\phi_{i,k,2,j,\tau },\ldots \phi_{i,k,t,j,\tau }\ldots ,\ldots \right]$$
(22)$$\phi_{j,\tau }=\left[\phi_{1,j,\tau },\phi_{2,j,\tau },\ldots ,\phi_{i,j,\tau },\ldots \right]$$
(23)$$\phi ={\left[\phi_{1,1}^{T},\phi_{1,2}^{T},\ldots ,\phi_{j,1}^{T},\phi_{j,2}^{T}\ldots ,\phi_{j,\tau }^{T},\ldots ,\ldots \right]}^{T}$$
where S is the vector consists of all considered time-dependent turning movements and ϕ is the corresponding proportion vector. $\phi_{i,j,\tau }$ is the vector consists of the proportions of users between all time-dependent paths of O–D pair i related to the turning movements at node j with departure time τ. $\phi_{j,\tau }$ is the vector consists of the proportions of users between all time-dependent paths of all O–D pairs related to the turning movements at node j with departure time τ.

Considering the error term, similarly, a linear relationship between the O–D demand vector D, the path flow vector F, and the turning movement vector S is assumed to be:(24)S=ϕF+η=ϕPD+η
where $\eta =\left(\eta_{1},\eta_{2},,\ldots \right)$ are mutually independent random variables with zero mean.

#### 4.1.4. O–D Demand, Path Flow, and Link Flow

Define $v_{j,\tau }$ as the flow of link j at time τ, and $\Psi_{i,k,t,j,\tau }$ as the proportion of users between path k of O–D pair i with departure time t choosing link j at time τ.

The time-dependent link flow can be derived from the time-dependent path flow as follows:(25)$$v_{j,\tau }=\underset{i}{\sum }\underset{k}{\sum }\underset{t}{\sum }\Psi_{i,k,t,j,\tau }f_{i,k,t}$$

Define vectors $\Psi_{i,j,\tau }$, $\Psi_{j,\tau }$, Ψ and V as follows:(26)$$V={\left[v_{1,1},v_{1,2},\ldots \underset{\text{time-dependent flow of link j}}{\underset{︸}{v_{j,1},v_{j,2},\ldots ,v_{j,\tau },\ldots }},\ldots \right]}^{T}$$
(27)$$\Psi_{i,j,\tau }=\left[\Psi_{i,1,1,j,\tau },\Psi_{i,1,2,j,\tau },\ldots \Psi_{i,k,1,j,\tau },\Psi_{i,k,2,j,\tau },\ldots ,\Psi_{i,k,t,j,\tau },\ldots ,\ldots \right]$$
(28)$$\Psi_{j,\tau }=\left[\Psi_{1,j,\tau },\Psi_{2,j,\tau },\ldots ,\Psi_{i,j,\tau },\ldots \right]$$
(29)$$\Psi ={\left[\Psi_{1,1}^{T},\Psi_{1,2}^{T},\ldots ,\Psi_{j,1}^{T},\Psi_{j,2}^{T},\ldots ,\Psi_{j,\tau },\ldots ,\ldots \right]}^{T}$$
where V is the vector consists of all considered time-dependent link flows and Ψ is the corresponding proportion vector. $\Psi_{i,j,\tau }$ is the vector consists of the proportions of users between all time-dependent paths of O–D pair i choosing link j with departure time τ. $\Psi_{j,\tau }$ is the vector consists of the proportions of users between all time-dependent paths of all O–D pairs choosing link j with departure time τ.

Considering the error term, a linear relationship between the O–D demand vector D, the path flow vector F, and the sub-path flow vector V is assumed to be:(30)V=ΨF+ξ=ΨPD+ξ
where $\xi =\left(\xi_{1},\xi_{2},\ldots \right)$ are mutually independent random variables with zero mean.

### 4.2. Updating Observed Information

According to Equations (11), (17), (24), and (30), the whole set of random variables considered in our model can be described by the linear expression:(31)$$\left(\begin{array}{c} D\\ F\\ F_{sub}\\ S\\ V \end{array}\right)=\left(\begin{array}{cccc} I & 0 & 0 & 0\\ P & 0 & 0 & 0\\ \varphi P & I & 0 & 0\\ \phi P & 0 & I & 0\\ \Psi P & 0 & 0 & I \end{array}\right)\left(\begin{array}{c} D\\ \varepsilon \\ \eta \\ \xi \end{array}\right)$$

Note that P, φ, ϕ, and Ψ can be obtained by solving the dynamic user equilibrium (DUE) problem based on the prior O–D demand. According to assumptions introduced in Section 2, if D are multivariate normal random variables with mean E(D) and variance $\Sigma_{D}$, the expected value (vector) of all random variables $E\left(D,F,F_{sub},S,V\right)$ is
(32)$$\left(E\begin{array}{c} E\left(D\right)\\ E\left(F\right)\\ \left(F_{sub}\right)\\ E\left(S\right)\\ E\left(V\right) \end{array}\right)=\left(\begin{array}{cccc} I & 0 & 0 & 0\\ P & 0 & 0 & 0\\ \varphi P & I & 0 & 0\\ \phi P & 0 & I & 0\\ \Psi P & 0 & 0 & I \end{array}\right)\left(\begin{array}{c} E\left(D\right)\\ E\left(\varepsilon \right)\\ E\left(\eta \right)\\ E\left(\xi \right) \end{array}\right)$$

The variance-covariance matrix $\Sigma_{\left(D,F,F_{sub},S,V\right)}$ is:(33)$$\Sigma_{\left(D,F,F_{sub},S,V\right)}=\left(\begin{array}{ccccc} \Sigma_{D} & \Sigma_{D}P^{T} & \Sigma_{D}{\left(\varphi P\right)}^{T} & \Sigma_{D}{\left(\phi P\right)}^{T} & \Sigma_{D}{\left(\Psi P\right)}^{T}\\ P\Sigma_{D} & P\Sigma_{D}P^{T} & P\Sigma_{D}{\left(\varphi P\right)}^{T} & P\Sigma_{D}{\left(\phi P\right)}^{T} & P\Sigma_{D}{\left(\Psi P\right)}^{T}\\ \varphi P\Sigma_{D} & \varphi P\Sigma_{D}P^{T} & \varphi P\Sigma_{D}{\left(\varphi P\right)}^{T}+D_{\varepsilon } & \varphi P\Sigma_{D}{\left(\phi P\right)}^{T} & \varphi P\Sigma_{D}{\left(\Psi P\right)}^{T}\\ \phi P\Sigma_{D} & \phi P\Sigma_{D}P^{T} & \phi P\Sigma_{D}{\left(\varphi P\right)}^{T} & \phi P\Sigma_{D}{\left(\phi P\right)}^{T}+D_{\eta } & \phi P\Sigma_{D}{\left(\Psi P\right)}^{T}\\ \Psi P\Sigma_{D} & \Psi P\Sigma_{D}P^{T} & \Psi P\Sigma_{D}{\left(\varphi P\right)}^{T} & \Psi P\Sigma_{D}{\left(\phi P\right)}^{T} & \Psi P\Sigma_{D}{\left(\Psi P\right)}^{T}+D_{\xi } \end{array}\right)$$
where $D_{\varepsilon }$, $D_{\eta }$, $D_{\xi }$ are the variance matrixes of ε, η and ξ respectively.

Given Equations (32) and (33), we can update the mean and the variance-covariance matrix of variables based on multiple sources of traffic counts using Equations (3) and (4). The observed variables include observed time-dependent O–D pair demand, path flows, sub-path flows, intersection turning movements, and link counts.

## 5. A Stepwise Algorithm

In a real-sized transportation network, the numbers of links, paths, and O–D pairs are usually very large, so the dimensions of some variance-covariance matrices in Equations (3) and (4) could be very large, which makes the proposed model difficult to solve. Specifically, if the matrix size is large, matrix inversion involved in Equations (3) and (4) is difficult to compute, and sometimes the matrix may not be reversible at all. To simplify the model, we propose to use a stepwise method to solve the Bayesian model as shown in Figure 1:

Step 1: Initialization. From historical data, we obtain the prior distribution of time-dependent O–D demand with mean E(D) and variance $\Sigma_{D}$. The variance $\Sigma_{D}$ is diag(αE(D)), where α is the coefficient of variation. Define n as the iteration number and set n=0.

Step 2: Given the mean E(D) of time-dependent O–D demand, obtain the choice proportion matrices P, φ, ϕ, and Ψ by solving the DUE problem.

Step 3: Calculate the mean and variance-covariance matrix of time-dependent O–D demand, path flows, sub-path flows, intersection turning movements, and link counts based on Equations (31) and (32).

Step 4: If the time-dependent sub-path travel time is observed without knowing sub-path flows, convert the observed sub-path travel time to time-dependent sub-path flow as follows. Otherwise go to Step 5.

Step 4.1: If the observed mean travel time of sub-path j with departure time $\tau_{d}$ is ${\bar{t}}_{j,\tau_{d}}$ and the prior total flow of sub-path j with departure time $\tau_{d}$ is $f_{sub,j,\tau_{d}}$ (obtained by Step 2), sample the normal distribution $N\left({\bar{t}}_{j,\tau_{d}},\lambda {\bar{t}}_{j,\tau_{d}}\right)$ for $f_{sub,j,\tau_{d}}$ iterations, and the sampled $f_{sub,j,\tau_{d}}$ random numbers are treated as the travel time for each user.

Step 4.2: The departure time at the start node of the considered sub-path for each user is the same since they are given, then by sorting out the sampled travel time for each user with the same arrival time at the tail node of the considered sub-path, several time-dependent sub-path flows can be derived and these flows can be treated as the observed variables and go to Step 5.

Step 5: Update the mean and variance-covariance matrices of time-dependent O–D demand, path flows, sub-path flows, intersection turning movements, and link counts based on the observed data using Equations (3) and (4).

Step 6: Convergence test. Set n=n+1. If $n=n^{max}$ or $\underset{i}{\sum }\underset{t}{\sum }{\left[E\left(d_{i,t}\right)-E\left(d_{i,t}^{*}\right)\right]}^{2}<\omega$ then stop, where $E\left(d_{i,t}^{*}\right)$ is the updated mean flow of O–D pair i with departure time t, ω is a small number to control convergence, and $n^{max}$ is the maximum iteration number. Return the updated mean and variance-covariance matrices of time-dependent O–D demand, path flows, sub-path flows, intersection turning movements, and link flows, based on which the point estimation and the corresponding probability intervals will be identified. Otherwise, continue to Step 7.

Step 7: Update the mean and variance of time-dependent O–D demand:(34)$$E\left(d_{i,t}\right)=\rho E\left(d_{i,t}^{*}\right)+\left(1-\rho \right)E\left(d_{i,t}\right)$$
(35)$$\Sigma_{D}=diag\left(\alpha E\left(D\right)\right)$$
where ρ, 0<ρ<1 is a relaxation factor. Then go to Step 2.

In this algorithm, given a seed (prior) O–D demand matrix, an updated O–D demand matrix can be obtained based on the Bayesian statistical method. In each iteration, the choice proportion matrices P, φ, ϕ, and Ψ are fixed. Since these proportion matrices vary with the O–D demand matrix, they will also be updated when an updated O–D demand matrix is derived. The algorithm goes to the next iteration based on the updated O–D demand matrix and choice proportion matrices. In Step 5, we update observed variables one by one. In this case, according to Equations (3) and (4), we do not need to calculate the inverse of a matrix because these matrices degenerates to column vectors or scalars. Specifically, $\Sigma_{\theta X}$, $\Sigma_{YZ}$,$\;\Sigma_{YX}$, $\Sigma_{XZ}$ are column vectors and $\Sigma_{XX}$ degenerates to a scalar. Then the number of calculations needed in Step 5 is linear in the dimensions of $E\left(D,F,F_{sub},S,V\right)$ and $\Sigma_{\left(D,F,F_{sub},S,V\right)}$. Because most calculations are involved in Step 5, the computational time of the proposed algorithm in each iteration is linear in the size of number of links, turning movements and sub-paths in the network.

## 6. Numerical Example

In this section, we demonstrate the proposed method and algorithm using Nguyen–Dupuis network as shown in Figure 2, which has been frequently used in the literature to verify methods related to transportation network modeling including the O–D demand estimation problem [29,34]. It consists of 13 nodes and 38 bidirectional links. Six nodes {12, 1, 4, 8, 2, 3} are terminal nodes, which could be either origins or destinations. Vehicles can travel from left to right (from origins {12, 1, 4} to destinations {8, 2, 3}) or from right to left (from origins {8, 2, 3} to destinations {12, 1, 4}). Therefore, in total there are 18 O–D pairs. The O–D matrix is time-dependent with 15-min intervals and the number of time intervals is six. Demand for each O–D pair at each departure time is 30 in the seed matrix. We suppose the “true” O–D demand matrix is known, which is generated from the seed matrix randomly. The observed data are assumed to be collected by sensors in the network. Specifically, we assign the “true” matrix in the network by DUE method and place sensors in the network to obtain the sensor data, which collect travel time on 4 sub-paths, time-dependent turning movements at 18 intersections and time-dependent link counts on 30 links. These sensor data, as tabulated in Table 2, Table 3 and Table 4, serve as our observed data. In such a manner the observed data are consistent with the “true” matrix and assignment method in the model. We then try to estimate time-dependent O–D demand reversely from the observed data to match the “true” matrix. The DUE method is a standalone procedure in the model, which can be solved by off-the-shelf traffic software. In this paper, we used a dynamic assignment and simulation model—DYNASMART-P 1.3.0—to solve DUE.

To measure the performance of the proposed Bayesian method and the algorithm, three aggregate measures were used: the percentage root-mean-square error (%RMSE), the mean absolute error (MAE) and Theil’s inequality coefficient U [58] for traffic counts, to measure the fit between estimated and observed traffic counts:(36)$$\mathrm{RMSE}=\frac{\sqrt{\frac{1}{N}\sum_{n=1}^{N}{\left(y_{n}^{est}-y_{n}^{obs}\right)}^{2}}}{\frac{1}{N}\sum_{n=1}^{N}y_{n}^{obs}}\times 100\%$$
(37)$$\mathrm{MAE}=\frac{1}{N}\overset{N}{\underset{n=1}{\sum }}\left|\left(y_{n}^{est}-y_{n}^{obs}\right)\right|$$
(38)$$U=\frac{\sqrt{\frac{1}{N}\sum_{n=1}^{N}{\left(y_{n}^{est}-y_{n}^{obs}\right)}^{2}}}{\sqrt{\frac{1}{N}\sum_{n=1}^{N}{\left(y_{n}^{est}\right)}^{2}}+\sqrt{\frac{1}{N}\sum_{n=1}^{N}{\left(y_{n}^{obs}\right)}^{2}}}$$
where N is the number of measurements, $y_{n}^{est}$ is the estimated measurement, and $y_{n}^{obs}$ is the observed measurement. Note that the value of U is between zero and one. U=0 implies a perfect fit between the estimated and observed measurements, while U=1 indicates the worst possible fit.

Similarly, to measure the fit between estimated and “true” O–D demand, three measures were used as following:(39)$$\mathrm{OD}\_\mathrm{RMSE}=\frac{\sqrt{\frac{1}{M}\sum_{m=1}^{M}{\left(d_{m}^{est}-d_{m}^{obs}\right)}^{2}}}{\frac{1}{M}\sum_{m=1}^{M}d_{m}^{obs}}\times 100\%$$
(40)$$\mathrm{MAE}=\frac{1}{N}\overset{N}{\underset{n=1}{\sum }}\left|\left(y_{n}^{est}-y_{n}^{obs}\right)\right|$$
(41)$$U=\frac{\sqrt{\frac{1}{N}\sum_{n=1}^{N}{\left(y_{n}^{est}-y_{n}^{obs}\right)}^{2}}}{\sqrt{\frac{1}{N}\sum_{n=1}^{N}{\left(y_{n}^{est}\right)}^{2}}+\sqrt{\frac{1}{N}\sum_{n=1}^{N}{\left(y_{n}^{obs}\right)}^{2}}}$$
where M is the number of O–D pairs, $d_{m}^{est}$ is the estimated O–D demand, and $d_{m}^{obs}$ is the “true” O–D demand.

Starting from the seed matrix and the sensor data, the time-dependent O–D demand is estimated by the procedure introduced in Section 5. The value of α in Step 1 is 0.5, λ in Step 4.1 is 1.0, and ρ in Step 7 is 0.1.

The total O–D variance is the sum of variance of each time dependent O–D demand. Figure 3 shows how the total O–D variance changes within one iteration after the traffic count is updated one by one. It shows that the total O–D variance is decreasing with each added and updated traffic count. Smaller variance means lower uncertainty in the estimation, so updating each traffic count can improve accuracy of O–D estimation. Figure 3 also indicates that more traffic counts can lead to lower variance of the dynamic O–D demand estimation, since more updated information can be used to improve the O–D estimation. Because traffic counts are updated one by one in the proposed algorithm, when new traffic data come in, there is no need to resolve the Bayesian statistical model from scratch, there is just the need to continue to update the procedure with the additional traffic data. In real-world applications, we can measure the quality of traffic data by analyzing the resultant variance of the dynamic O–D demand estimation, so as to determine whether to add additional traffic data in the procedure or not.

Figure 4 illustrates how the measures of traffic count performance change at each iteration. It shows that three measures of performance have similar trends and they are all decreasing by iterations (although there are small fluctuations). This indicates that the proposed method normally can identify a solution that reduces the total error of traffic counts compared to that of last iteration. There are 30 iterations shown in Figure 4, till which the three measures of performance become flat. Noticeably, after 30 iterations, %RMSE has been reduced from around 36% to 6%, MAE has been reduced from around 51.00 to 7.00, and Theil’s inequality coefficient U has been reduced substantially from the initial value 0.25 to 0.035. These results demonstrate the high quality of the proposed Bayesian method for dynamic O–D demand estimation.

Table 2, Table 3 and Table 4 show the relative errors between the estimated and observed sub-path travel time, node turning movements, and link flows respectively. Note that the relative errors of almost 50% traffic counts are less than 5%, and the relative errors of 85% traffic counts are less than 10%. Note that relative errors of traffic counts with high values are small. Specifically, for the sub-path travel time, the relative errors are all small and less than 5%. For the node turning movement, relative errors of about 80% estimation are less than 10%. For the link counts, about 80% of the relative errors are less than 10% and over 50% of them are less than 5%. These results indicate that not only is the total error of the estimation obtained by the proposed Bayesian method reduced significantly (as also shown by Figure 4), but also the errors for each type of traffic counts are small. This demonstrates that the proposed Bayesian method can synthesize multiple sources of data well and provide good estimation.

In the literature, some methods used to estimate dynamic O–D demand can also have high estimation accuracy [36,37,38], and the accuracy is normally related to the error between the “true” and estimated O–D demand. However, traffic flows including the “true” O–D demand normally have high uncertainty in reality, and these methods cannot provide estimation information related to the uncertainty. As a statistical method, the proposed method can provide not only the point estimates (i.e., expected values), but also the variances, which represent the associated uncertainty for the corresponding O–D demand. Based on the expected values and variances, we can obtain the posterior distribution of the time-dependent O–D demand. According to the posterior distribution, the confidence intervals for each O–D demand can be identified. In summary, the proposed Bayesian statistical method can provide not only point estimates of dynamic O–D demand with high accuracy, but also the corresponding statistical information to capture the uncertainty and improve the reliability of the estimates.

Based on the posterior distribution of the time-dependent O–D demand, Figure 5 shows the 95% confidence intervals for each time-dependent O–D demand estimates. Note that because most variances are small, the lengths of the 95% confidence intervals are also small, which means low uncertainty involved in the estimated O–D demand. It can be seen that the length of intervals for some O–D demand estimates are much smaller than lengths of others. This is because according to the traffic assignment proportions of traffic counts (i.e., φ, ϕ, and Ψ in Equation (31)), some O–D pairs have much more traffic counts related to them. In such a case, the corresponding variance and length of confidence interval could be small since there is more information to update them and reduce the variability. In fact, if we have more observed traffic counts, the variances will be even smaller and the resultant O–D demand estimates will have even lower uncertainty (as demonstrated by Figure 3). This gives a hint to determine which links, nodes, and/or paths need to be observed when estimating traffic flows by the Bayesian method; that is, the network sensor location problem. We can locate sensors on a set of links, nodes, and/or paths which lead to lower uncertainty of the dynamic O–D demand estimation [51].

Table 5 shows the values of three measures (as shown in Equations (39)–(41)) related to the performance of O–D demand estimates. According to the three measures, it can be seen that the total error between the estimated and “true” O–D demands is relatively small. For example, the resultant OD_U is around 0.1. It also can be seen that the errors between the estimated and “true” O–D demands in some time intervals are larger than errors in other time intervals. This is because in those time intervals with larger errors, only a few traffic counts are related to the O–D pairs in the considered time intervals. Thus, the precision of the O–D demand estimates is much lower.

To further study the impact of the number of traffic counts on the precision of the O–D demand estimates, take the O–D demand estimates in time interval 2 for example. Table 6 compares performance of O–D demand estimates with different number of traffic counts related to the considered time interval. For the purpose of comparison, only traffic counts on links are considered in Table 6. When users from an O–D pair in time interval 2 use a link collecting traffic counts, traffic count on this link is treated as related to the O–D pair in time interval 2. From Table 6, it can be seen that when the number of related traffic counts increases, the error between the estimated and “true” O–D demands decreases, as demonstrated by the changes of the three measures’ values. Thus, if we have more observed traffic counts, the errors will be even smaller and the resultant O–D demand estimates have even higher accuracy, as was highly expected. In summary, according to Figure 5 and Table 6, more observed traffic counts can lead to lower uncertainty and higher precision of O–D demand estimates.

## 7. Conclusions

To estimate dynamic O–D demand, a Bayesian method was proposed in this paper. The method can synthesize multiple sources of data, including link counts, turning movements at intersections, sub-path flows, and/or sub-path travel time. Demand between each O–D pair at each departure time is assumed to satisfy normal distribution. By solving the DUE problem based on a prior O–D demand, the prior distribution of the considered variables can be obtained, including a vector of expected values and variance-covariance matrix. The relationships among all sources of data and all time dependent O–D demands can be established by variance-covariance matrices. By updating traffic counts, the posterior distribution of all variables can be built based on the prior distribution. According to the posterior distribution, both point estimation and the corresponding confidence intervals can be identified. The connection among link counts, turning movements at intersections, sub-path flows, and O–D demand can be obtained by the corresponding traffic assignment proportions. The connection between sub-path travel time and O–D demand cannot be established directly, therefore we convert each observed sub-path travel time to several sub-path flows to incorporate the travel time measurement in dynamic O–D estimation.

Updating the traffic count usually requires computing matrix inversion, which is rather involved even on a network with a modest size. To avoid matrix inversion, a stepwise algorithm is developed for solving the proposed Bayesian method. In this algorithm, the traffic count is updated one by one in each iteration. Thus, matrix inversions are avoided since matrices in the proposed Bayesian statistical model degenerate to column vectors or even scalars. Moreover, since we update a traffic count at a time, when additional traffic counts are observed, we just need to proceed to update the extra traffic counts and do not need to resolve the Bayesian statistical model from the scratch.

The results of the numerical example based on the Nguyen–Dupuis network shows that the total variability of O–D demand decreases with each added traffic count. More traffic counts can lead to smaller variance of the dynamic O–D demand, which means updating each traffic count can reduce the uncertainty in the O–D estimation. Using the proposed algorithm, the total deviations between estimated and observed traffic counts decreases at each iteration, as supported by three measures of performance. After a few iterations, the three measures all decrease to small values and become flat, which implies a good fit between estimated and observed traffic counts. Moreover, the source-specific deviations between estimated and observed traffic counts are small too. This demonstrates the proposed Bayesian method can synthesize multiple sources of data well. It also implies that more traffic counts lead to lower uncertainty of the O–D demand estimates, resulting in a more accurate estimation.

Although normal distribution assumption of traffic flows is reasonable in some cases, sometimes this assumption is not in accordance with reality, especially when the traffic volume is low. Thus, in future research, it is worthwhile to relax the multivariate normal distribution assumption for the prior time-dependent O–D demands. Moreover, in the proposed Bayesian statistical model, the method of considering sub-path travel times is an approximate method. Hence, it is worthwhile to further investigate a more effective method in order to incorporate sub-path travel times in dynamic O–D demand estimation. Last but not least, additional experiments can be performed to evaluate the performance of the proposed model and algorithm in various larger or real transportation networks.

## Figures and Tables

**Figure 1 sensors-21-04971-f001:**
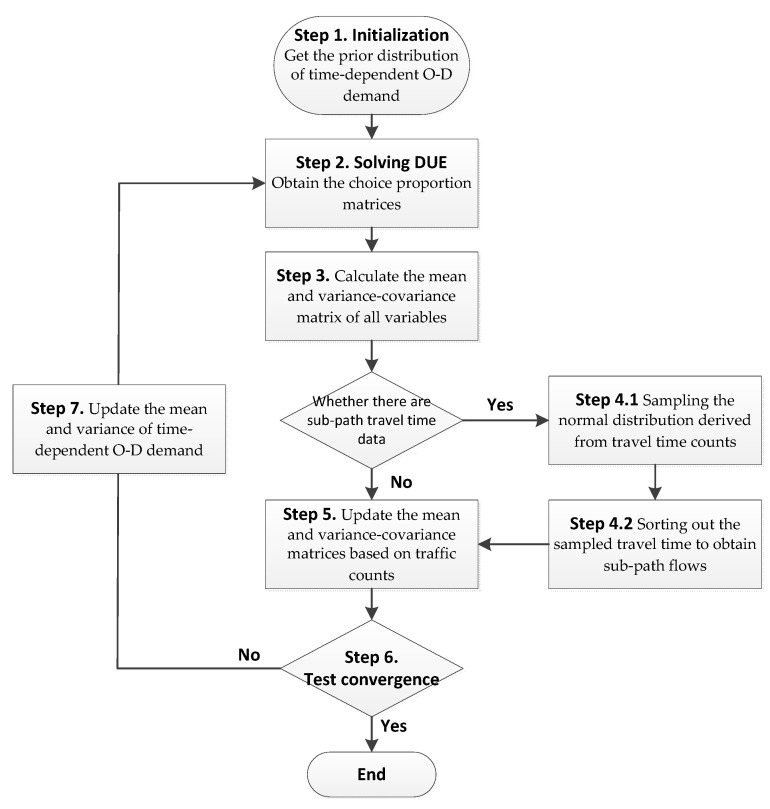
The stepwise method to solve the Bayesian model.

**Figure 2 sensors-21-04971-f002:**
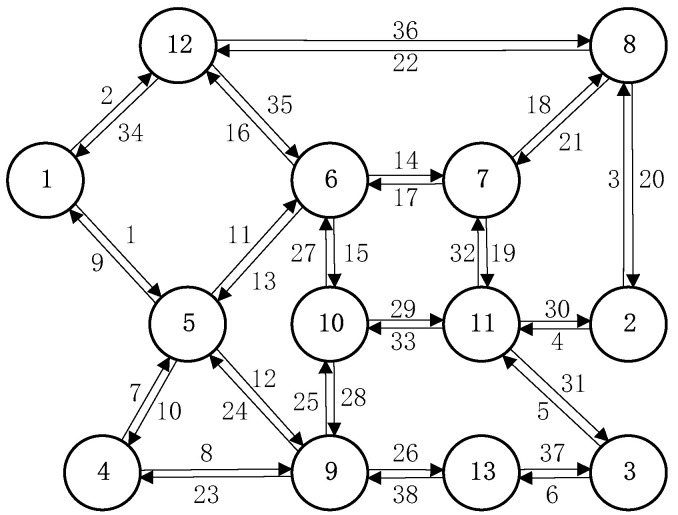
The Nguyen–Dupuis network.

**Figure 3 sensors-21-04971-f003:**
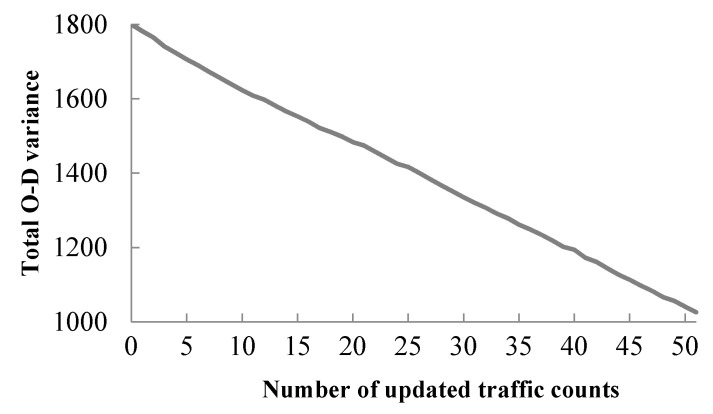
Total O–D variance after updating each traffic count in one iteration.

**Figure 4 sensors-21-04971-f004:**
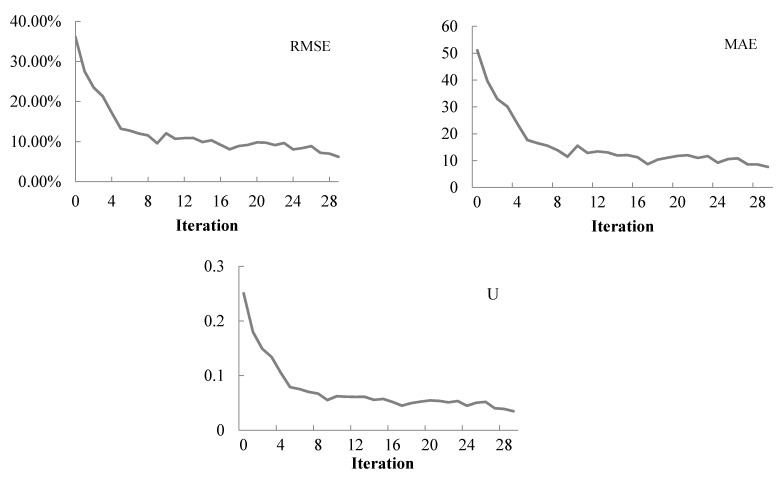
Measures of performance after each iteration.

**Figure 5 sensors-21-04971-f005:**
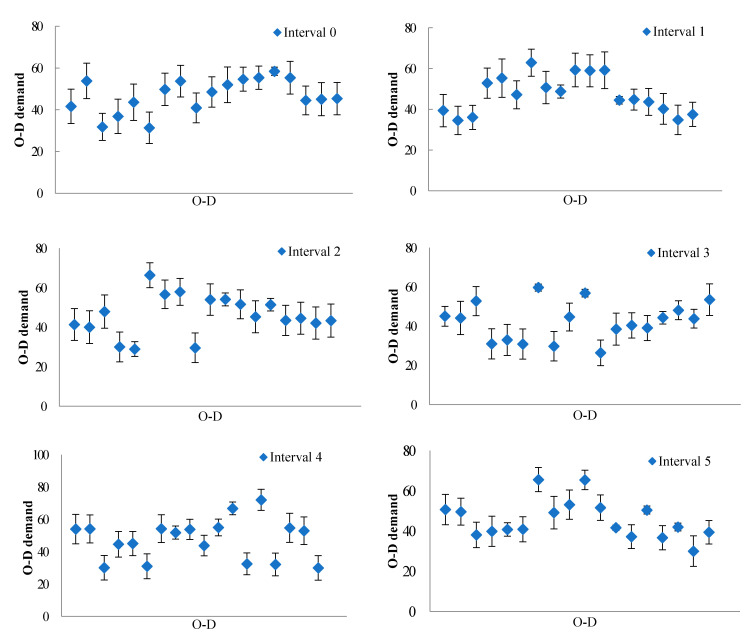
Confidence intervals of 95% for O–D demand estimates.

**Table 1 sensors-21-04971-t001:** Studies on O–D demand estimation using multiple sources of data.

References	Case	Traffic Counts					Variability Information
		Link Flow	Node Turning Movements	Partial Path Flow	Link Travel Time	Path Travel Time	
[31]	Dynamic	×	√	×	×	×	×
[32,33]	Dynamic	√	√	×	×	×	×
[34]	Static	√	×	√	×	×	√
[35]	Static	√	×	√	×	×	×
[36,37,38,39]	Dynamic	√	×	√	×	×	×
[40]	Dynamic	√	×	×	√	×	×
[42]	Dynamic	√	×	×	×	√	×
This paper	Dynamic	√	√	√	×	√	√

**Table 2 sensors-21-04971-t002:** Observed and estimated sub-path travel time.

Sub-Path	Departure Time	Observed	Estimated	Relative Error (%)	Sub-Path	Departure Time	Observed	Estimated	Relative Error (%)
5-6-7-8	1	7.5	7.45	0.67	5-9-10-11	2	7.5	7.34	2.13
5-6-7-8	3	5	4.83	3.4	5-9-10-11	4	7	7.1	1.43

**Table 3 sensors-21-04971-t003:** Observed and estimated node turning movements.

Turning Movement	Departure Time	Observed	Estimated	Relative Error (%)	Turning Movement	Departure Time	Observed	Estimated	Relative Error (%)
2-11-10	0	63	68	8.00	7-6-5	3	36	27	25.00
10-6-12	1	41	51.21	24.90	7-6-12	4	40.72	44	8.06
3-11-10	1	91	82.07	9.81	7-6-5	4	127.26	120.02	5.69
2-11-10	1	124	132	6.45	2-11-7	4	57.92	63	8.77
7-6-5	2	39	40	2.56	2-11-10	4	73.42	74.34	1.25
7-11-3	2	50	47	6.00	7-6-12	5	70.86	75.7	6.83
3-11-10	2	35	32.67	6.66	7-6-5	5	108.25	105.87	2.20
2-11-10	2	138	141	2.17	2-11-10	5	52	53.66	3.19
10-6-12	3	35	36	2.86	7-6-12	6	34.63	32.75	5.43

**Table 4 sensors-21-04971-t004:** Observed and estimated link flows.

Link	Departure Time	Observed	Estimated	Relative Error (%)	Link	Departure Time	Observed	Estimated	Relative Error (%)
8-12	0	143.06	138.28	3.34	3-13	3	140.09	147.18	5.06
5-9	0	161	163.11	1.31	3-13	4	142.35	150.56	5.77
3-13	0	139	163.42	17.57	5-6	4	140	142	1.43
4-5	0	137	146.5	6.93	6-5	4	179.26	176.59	1.49
12-8	1	154.26	161.94	4.98	7-8	4	167.25	153	8.52
8-12	1	189.81	192.39	1.36	8-7	5	159.23	166.01	4.26
5-9	1	285	282.8	0.08	12-8	5	216.47	194.13	10.32
9-5	1	240	260.61	8.59	8-12	5	186.88	189.59	1.45
8-12	2	147.23	147.96	0.50	1-12	5	195.59	172.38	11.87
12-1	2	136	160.04	17.68	3-13	6	93.34	96.43	3.31
5-9	2	339.94	337.66	0.67	12-8	6	141.67	121.21	14.44
9-5	2	322.75	311.92	3.35	8-12	6	128.86	138.79	7.71
1-12	3	99.67	88.73	10.98	12-1	6	107.41	111.69	3.99
5-9	3	284.45	309.13	8.68	5-6	6	21	25	19.05
9-5	3	100.96	97.64	3.29	9-13	6	35.11	34	3.16

**Table 5 sensors-21-04971-t005:** Comparison between estimated and true O–D demand.

Time Interval	%OD_RMSE	OD_MAE	OD_U
0	22.17%	8.36	0.11
1	34.97%	13.00	0.18
2	31.42%	11.53	0.16
3	29.22%	11.77	0.15
4	15.07%	5.53	0.07
5	23.20%	9.29	0.11
Total	25.30%	9.42	0.12

**Table 6 sensors-21-04971-t006:** Performance of O–D demand estimates with different number of traffic counts (time interval = 2).

Number of Related Traffic Counts	%OD_RMSE	OD_MAE	OD_U
4	33.69%	12.87	0.18
8	31.55%	11.66	0.17
12	21.04%	8.09	0.11
16	18.95%	7.76	0.10

## Data Availability

The data presented in this study are available on request from the corresponding author.

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
