# Peer review of "A Bayesian Method for Dynamic Origin–Destination Demand Estimation Synthesizing Multiple Sources of Data"

_sensors, 2021, doi:10.3390/s21154971_

Round 1

Reviewer 1 Report

In this paper, a Bayesian method for dynamic origin-destination demand estimation synthesizing multiple sources of data is presented. A numerical example is performed to demonstrate the effectiveness of the presented Bayesian method and solution algorithm. This paper is informative and well written. The reviewer recommends publication of this manuscript but the authors should first make changes to address the following points:

1. Section of Bayesian Statistical Method: it is unclear how to synthesize multiple sources of data using the presented Bayesian theory.
2. Eqs. (36) and (39), it looks strange placing “%” before the equation.
3. Section of Numerical Example: Is it possible to employ real data to validate the performance of the presented method?
4. The origin-destination demand estimation should be a widely-studied problem in Bayesian network. The authors should include these references in Section of Literature Review. In addition, the author should include some references on recent development of Bayesian system identification, for example:                                                   
Huang, C.S. Shao, B. Wu, J. L. Beck, Hui Li, State-of-the-art review on Bayesian inference in structural system identification and damage assessment, Advances in Structural Engineering,2019, 22(6): 1329-1351
J.L. Beck, Bayesian system identification based on probability logic, Struct. Control. HealthMonit. 17(7) (2010) 825–847.
P.L.Green,E.J.Cross, K.Worden,Bayesian system identification of dynamical systems using highly informative training data,Mechanical Systems and Signal Processing, 2015, 56–57: 109-122

Reviewer 2 Report

In the paper, a Bayesian method was proposed to estimate dynamic O-D demand. The method can synthesize multiple sources of data, including link counts, turning movements at intersections, sub-path flows, and/or sub-path travel time. A numerical example is conducted on Nguyen–Dupuis network to demonstrate the effectiveness of the proposed approach. The results are encouraging. However, there are some issues that the author should consider to improve the structure and quality of the paper:
1.    In the abstract, the authors assert “It concludes that the proposed Bayesian method can effectively synthesize multiple sources of data and estimate dynamic O-D demands with fine accuracy”. The readers need to know from the beginning what is the accuracy. Thus, the numerical values could be given.
2.    In the first part of the paper, the authors present various approaches from the literature. Maybe, a synthesis of the solutions proposed in the literature depending on the type of analysis, which to highlight more clearly the advantages and disadvantages, is useful for readers. This synthesis can be given as a table.
3.    The flowchart of the stepwise algorithm could be introduced for an easy follow-up of the steps.
4.    The results must be compared with other methods proposed in the literature (presented in the first part of the paper) to highlight the accuracy of the proposed approach. 

Reviewer 3 Report

Dear authors,

I must say that I am very interested in your scientific paper. In my opinion, the paper is well written, the chosen area of research is highly needed and could be used to improve organization of traffic flow and its behavior prediction. I have only one recommendation: please check you spelling in the paper, there are some misspelling f.e. p. 9 line 352 (steowise/stepwise), p. 13 lines 458,459, etc.

Thank you for your contribution.

Best regards,

Reviewer

Reviewer 4 Report

Page 1; Please note that “and Tianpei Tang1” is written using different fonts.

Page 1, lines 30-31, “Intelligent Transportation Systems (ITS) have been vigorously developed by many cities around the world.”: My suggestion is to replace “developed by” with “implemented in”.

Page 1, lines 33-34, “O-D demand matrices can be obtained either from household surveys or estimated by traffic counts.”: My suggestion is to replace “or” with “or/and” since many times traffic counts are used to validate the O-D results obtained through household-based questionnaire surveys.

Page2, lines 46-47, “According to the characteristic of data collected, sensors in this paper are categorized as follows.”: Since cost plays a vital role in the selection of the category of sensors, please provide, within the manuscript, some estimates concerning the cost of each category of sensors.

Page 2, lines 62-64, “However, travel time data can often be collected much more easily (e.g., by Vehicle-ID sensors) than the volume data (especially along a path or a partial path).”: I am not sure if it is easier to collect “travel time” data than “traffic volume” data. Please analytically justify your statement.

Page 2, Section 2. Literature review: I appreciate the way you organize your information in the specific section. However, I would suggest adding a Table at the end of the specific Section to summarize the advantages and disadvantages of the methods presented.

Page 4, line 151: Please change “Image/video sensors.” to “image/video sensors.”.

Page 10, lines 400-401: You use the Nguyen–Dupuis network for the needs of the numerical example. Please justify, within the manuscript, the selection of the specific test network compared to other alternatives (e.g., Sioux Falls network).

Page 13, “Fig. 3 measures of performance after each iteration”: Please change the specific heading to “Fig. 3 Measures of performance after each iteration”.

Page 15, lines 517-519, “Thus, if we have more observed traffic counts, the errors will be even smaller and the resultant O-D demand estimates have even higher accuracy.” I would suggest adding “as it was highly expected” at the end of the above-mentioned sentence.

Page 16, Section 7. Conclusions: Please include the constraints and limitations of your research. It would also be worthwhile to refer to the results of similar case studies worldwide, if any, and to discuss the compliance of their findings with your findings.

Round 2

Reviewer 2 Report

The authors have tried and largely succeeded in removing any doubts raised by the reviewer.